# Identification of *C. elegans* ASNA-1 domains and tissue requirements that differentially influence platinum sensitivity and growth control

**Dorota Raj[1], Agnieszka Podraza-Farhanieh[1], Pablo Gallego[2¤], Gautam Kao[1]\*,
Peter Naredi[1,3]\***

**1** Department of Surgery, Institute of Clinical Sciences, Sahlgrenska Academy, University of Gothenburg, Gothenburg, Sweden, **2** Department of Medical Biochemistry and Cell Biology, University of Gothenburg, Gothenburg, Sweden, **3** Department of Surgery, Sahlgrenska University Hospital, Gothenburg, Sweden

¤ Current address: Discovery Sciences, BioPharmaceuticals R&D, AstraZeneca, Gothenburg, Sweden
\* gautam.kao@gu.se (GK); peter.naredi@gu.se (PN)

**Data Availability Statement:** All relevant data are within the manuscript and its Supporting Information files.

## Abstract

ASNA1 plays an essential role in cisplatin chemotherapy response, type 2 diabetes, and heart disease. It is also an important biomarker in the treatment response of many diseases. Biochemically, ASNA1 has two mutually exclusive redox-modulated roles: a tail-anchored protein (TAP) targeting function in the reduced state and a holdase/chaperone function in the oxidized state. Assigning biochemical roles of mammalian ASNA1 to biomedical functions is crucial for successful therapy development. Our previous work showed the relevance of the *C. elegans* ASNA-1 homolog in modeling cisplatin response and insulin secretion. Here we analyzed two-point mutants in highly conserved residues in *C. elegans* ASNA-1 and determined their importance in separating the cisplatin response function from its roles in insulin secretion. *asna-1(ΔHis164)* and *asna-1(A63V) point mutants*, which both preferentially exist in the oxidized state, displayed cisplatin sensitivity phenotype as well as TAP insertion defect but not an insulin secretion defect. Further, using targeted depletion we analyzed the tissue requirements of *asna-1* for *C. elegans* growth and development. Somatic depletion of ASNA-1 as well as simultaneous depletion of ASNA-1 in neurons and intestines resulted in an L1 arrest. We concluded that, targeting single residues in ASNA-1 affecting Switch I/Switch II domain function, in comparison to complete knockdown counteracted cisplatin resistance without jeopardizing other important biological functions. Taken together, our study shows that effects on health caused by ASNA1 mutations can have different biochemical bases.

## Author summary

Cisplatin is a chemotherapeutic widely used for cancer treatment. However, tumor resistance to cisplatin and its cytotoxic effects on healthy kidneys and neurons limits its use.

**Funding:** The work was supported by grants from the Swedish Cancer Society (https://www.cancerfonden.se/forskning) CAN 2018/664 (P.N.) and ALF Västra Götaland (https://www.alfvastragotaland.se/) nr: ALFGBG-722971 (P.N.); Stiftelsen Assar Gabrielssons Fond (https://www.agfond.se/) FB19-44 (DR) and Stiftelsen Assar Gabrielssons (https://www.agfond.se/) Fond FB20-32 (DR). The funders had no role in study design, data collection and analysis, decision to publish, or preparation of the manuscript.

**Competing interests:** The authors have declared that no competing interests exist.

The ability to reverse resistance and limit side effects could significantly improve cisplatin therapy in patients. To enhance the efficiency of cisplatin-mediated death more information on the molecular mechanisms and targets is needed. We study the targeting of *C. elegans* ASNA-1 protein as a way to increase cisplatin sensitivity. Human and *C. elegans* proteins are highly homologous. Knockdown of human or *C. elegans* ASNA1/ASNA-1 protein leads not only to cisplatin sensitivity phenotype but simultaneously results in decreased insulin secretion and type 2 diabetes in mice. Therefore, there is a need to dissect the clinically relevant functions of ASNA1/ASNA-1 in order to increase the sensitivity of resistant tumors without causing diabetes. Here we show that dissection of these two relevant functions (cisplatin response and insulin secretion) is possible. Moreover, we find a potential drug target in the protein that will allow us to re-sensitize resistant tumors to the cisplatin treatment while leaving insulin secretion undisturbed.

## Introduction

*C. elegans* ASNA-1 /*S. cerevisiae* GET3 /mammalian ASNA1 are essential cytosolic ATPases that are evolutionarily related to the bacterial arsenite transport factor ArsA [1], however the function in eukaryotes has clearly separated [2]. The mammalian homolog ASNA1 not only plays an essential role in tail-anchored (TA) protein insertion [3] and insulin secretion [4] but has been characterized as an important biomarker in treatment response in schizophrenia [5], predicting disease severity of dengue virus infection [6], differentiation between subjects with active and latent tuberculosis [7], a biomarker for abnormalities associated with ultra-high-risk for psychosis [8], as well as a biomarker for Down's syndrome [9]. Mutations in ASNA1 are associated with pediatric cardiomyopathy [10] and mutations in its pathway protein calcium modulating ligand (CAML) are linked in patients to hypotonia, brain abnormalities, and epilepsy [11]. Further ASNA1 interacts with and likely modulates the function of vesicle-associated protein B (VAPB) which is mutated in cases of amyotrophic lateral sclerosis [12]. Therefore, it is clear that ASNA1 has a role in a variety of human diseases and a deeper understanding of clinically relevant functions of ASNA1 is of general interest.

The diverse roles of ASNA1 in different disease manifestations are reflected in the different biochemical roles of this protein. The best understood canonical and most intensively studied role of ASNA-1/GET3/ASNA1 is in the endoplasmic reticulum (ER) targeting of a special class of membrane proteins called tail-anchored proteins (TAP) [13]. This targeting function strictly requires its ATPase activity [14] and the transmembrane regions of TAP proteins associate with the hydrophobic groove in dimeric ASNA1 before their insertion into the ER membrane. Insertion of TAPs via ASNA1/GET3 requires the ER membrane proteins WRB/CAML (in mammals) or Get1/Get2 (in yeast) [15] to which ASNA1/GET3 hands off the bound TAPs.

The ATPase activity of the ASNA-1/GET3 protein requires the function of the Switch I and Switch II domains. These two domains bind ATP and form so called 'loaded spring' that is released upon ATP hydrolysis [16,17]. The Switch II region collapses after ATP hydrolysis causing disruption in the network of conserved cross-monomer interactions that stabilize the composite groove [14]. In addition, the Switch II domain in yeast Get3 is essential for the transition from an open to a closed state and functions to assure structural changes that are induced by ATP or TAP binding [14]. This structural change ensures TAP binding to GET3 [18–20]. Structurally, residues in Switch I and Switch II domains of GET3 coordinate water and $Mg^{2+}$ molecules important for the ATPase activity [14]. The interaction between Switch I and Switch II domains is essential because an Asp57Asn mutant in the Switch I domain of

GET3 is defective for ATP hydrolysis and unable to rescue the hygromycin-sensitive phenotype of *Δget3* yeast [14].

However, it is clear that not all functions of ASNA1 and GET3 are a consequence of their canonical TAP targeting function. There are alternative roles for ASNA1/GET3 where it acts independently of its TAP insertion function and for these functions it thus does not require interaction with Get1/Get2 or WRB/CAML. Mammalian ASNA1 interacts with the tail-anchored protein, VAPB, but only via the FFAT motif in the MSP domain and not its hydrophobic groove which is essential for its interaction with tail-anchored proteins. Hence mammalian ASNA1 can display novel interaction patterns even with a tail-anchored protein that does not involve the canonical ASNA1/WRB/CAML pathway. VAPB may also work as an alternative receptor for ASNA1 instead of WRB/CAML [12]. Importantly for our understanding of ASNA1/GET3 functions, elegant biochemical and cell biological studies show upon oxidation-induced structural rearrangements GET3 acts as a holdase chaperone that binds to aggregated proteins under conditions of oxidative stress and ATP depletion [21,22]. Oxidation of conserved cysteines in Get3 converts the dimeric GET3 to a tetramer. This structural change masks the hydrophobic domain required for TA protein binding and substantially reduces its ATPase activity: Both the TA protein binding and ATPase properties of GET3, which are essential for its role in TAP targeting and interaction with WRB/GET1, are completely dispensable for the chaperone function of oxidized Get3. These lines of evidence demonstrate that under conditions of oxidative stress, Get3 promotes functions that do not require WRB/GET1. Moreover, *Voth et al.*, 2014 show that oxidized GET3 also acts as a general chaperone that can refold denatured citrate synthase in vitro in the absence of WRB/GET1 [22]. Moreover, GET3 directly binds to non-tail anchored proteins such as the chloride transporter Gef1p [23] and to the Gα subunit Gpa1 to act as a guanine nucleotide exchange factor [24]. In both cases, the binding is biologically meaningful since it leads to altered biochemical outcomes that are independent of GET1.

Our previous work on *C. elegans* ASNA-1 showed that ASNA-1 promotes insulin secretion and led later to the findings that insulin secretion is defective in ASNA1 knockdown mice leading to type 2 diabetes [25,26]. Further, the role of ASNA1 in the cisplatin sensitivity of human tumor cells was also modeled successfully in worms, and importantly the human ASNA1 gene can replace the worm gene for both cisplatin and insulin secretion functions [25,27]. Our previous work in worms demonstrates the TAP targeting function of ASNA-1 and shows that cisplatin exposure negatively impacts only targeting of the ASNA-1-dependent tail-anchored protein SEC-61β to the ER while having no effect on ASNA-1 independent TAPs [28].

Like the yeast ortholog, worm ASNA-1 is also oxidized via the two conserved cysteines. We find that the relevant roles of ASNA-1 in insulin secretion and protection from cisplatin toxicity can be separated based on the oxidative state of the protein. As with GET3, ASNA-1 exists in two redox-sensitive states: reduced (ASNA-1$^{RED}$) and oxidized (ASNA-1$^{OX}$) [28]. The single point mutant *asna-1(ΔHis164)*, in which the protein preferentially exists in the oxidized state, is not only as sensitive to cisplatin treatment as a complete deletion mutant but also has a severe defect in TAP insertion [27,28] showing that separation of function mutations exists and indicating that ASNA-1 function in insulin secretion is most likely independent of its role in TAP targeting and cisplatin response [27,28].

Here, we examined the expression of ASNA-1 and performed selective depletion to identify the tissue requirements for growth promotion. We also expanded our analysis of the *asna-1 (ΔHis164)* worms that have a lesion in the Switch II domain. Here we showed that this mutant, which has high levels of oxidized ASNA-1, has normal levels of insulin secretion and function. Further, we tested a set of viable and homozygous *C. elegans asna-1* point mutants to explore the possibility of completely separating clinically relevant functions of *asna-1* with genetic

variants. As a result, we characterized the highly conserved alanine in position 63 adjacent to Switch I as a relevant target for function separation. *asna-1(A63V)* mutants were sensitive to cisplatin but did not display any morphology, growth, lifespan, brood size, or oxidative stress phenotypes that were shown by *asna-1* deletion mutants. All these features made *asna-1 (A63V)* animals an excellent model to study the role of *asna-1* in the response to cisplatin without the compounding effects of other phenotypic defects. Indeed, *asna-1(A63V)* mutants showed cisplatin sensitivity defect and TAP insertion defect that was as severe as that seen in deletion mutants without any detectable defect in insulin secretion or signaling pathways. Moreover, this mutation led to higher levels of the oxidized form of the protein and altered its subcellular distribution. Lastly, we performed *in silico* modeling to better understand the consequences of the structural changes caused by the single-point mutations. For a multifunctional protein with non-overlapping functions, drugs affecting one function while not perturbing the other functions would reduce the undesirable side effects and might increase the chemotherapeutic response.

## Results

### ASNA-1 is broadly expressed in *C. elegans*

Expression from a transgene expressing ASNA-1::GFP (*svIs56*) was seen in sensory neurons, the intestine, and the hypodermis [25]. Taking advantage of the CRISPR/Cas9 technique we obtained a strain where two tags, mNeonGreen (mNG) and the auxin-inducible degron (AID), were inserted into the genomic locus of ASNA-1 at the C-terminus of the protein (ASNA-1::mNG::AID; *syb2249*; strain PHX2249). The mNeonGreen tag allowed us to more precisely characterize the ASNA-1 expression pattern in the context of all upstream and downstream elements. The mNeonGreen signal was visible strongly in the pharynx, intestine, muscles, proximal and distal germline cells, oocytes, developing vulva, vulval muscles, spermatheca, sperm, sheath cells, the most proximal pair of gonadal sheath cells (**Fig 1**), and in head neurons (**S1 Fig**). This analysis not only confirmed our previous observations [25] but also characterized previously unknown ASNA-1 expression sites in the *C. elegans* reproductive system. We concluded that the germline and somatic gonad are likely important places for ASNA-1 function given the broad expression of ASNA-1 in these tissues. To more precisely identify the cells expressing ASNA-1 in each tissue we next examined a bi-cistronic ASNA-1^SL2::mNeonGreen::H2B allele *syb5730* of the gene (strain PHX5730). From this allele, the ASNA-1 mRNA independently makes untagged ASNA-1 and nuclear-localized mNeonGreen to report on each nucleus that expresses the *asna*-1 mRNA. Analysis of the mNeonGreen signal in *asna-1(syb5730)* animals showed that virtually every cell in the body expresses the ASNA-1 mRNA throughout development in both the soma and the germline (**S2 Fig**).

### Soma-specific *asna-1* knockdown leads to L1 larval arrest

Worms lacking maternal and zygotic *asna-1* arrest at the 1st larval (L1) stage even in the presence of food [25]. L1 arrest is a characteristic of wild-type larvae hatched in the absence of food or of strong *daf-2*/insulin receptor mutants [29]. Taking into consideration how broadly ASNA-1 is expressed in somatic and germline tissues, we asked whether somatic depletion of ASNA-1::mNG::AID would cause a larval arrest phenotype similar to that seen after complete *asna-1* genetic depletion. *Arabidopsis* TIR1 is an F-box protein that interacts with the AID tag in the presence of auxin to assemble an E3 ubiquitin ligase complex and targets AID-containing proteins for rapid degradation by the proteasome [30,31]. Tissue or cell-type-specific expression of TIR1 limits the degradation of AID tagged proteins to TIR1-expressing cells. Taking the advantage of the auxin-inducible tissue-specific knockdown system, we created the

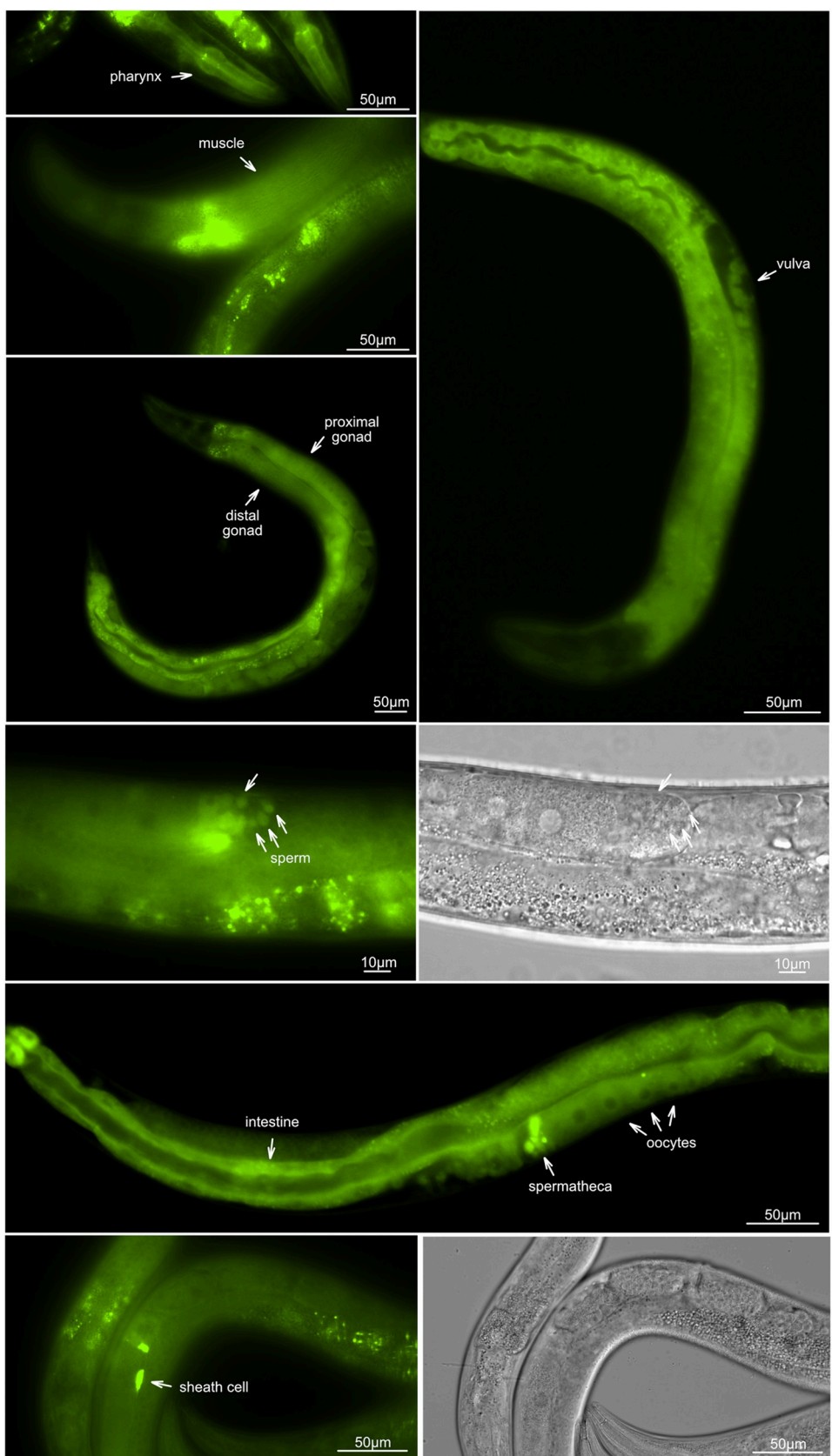

**Fig 1. ASNA-1 is broadly expressed in *C. elegans*.** Representative fluorescence and differential interference contrast (DIC) images worms expressing ASNA-1::mNeonGeen::AID (*syb2249*). White arrows indicate different structures where ASNA-1::mNeonGreen::AID is present.

strain expressing ASNA-1::mNG::AID (*syb2249*) and TIR1 pan-somatic driver *ieSi57*. 4th larval stage (L4) hermaphrodites were exposed to 1mM auxin (AUX) for 48h while they produced progeny and then removed from the plate. Their progeny, which remained on the auxin-containing plates, were analyzed 48 hours after the removal of the mothers (**S3A Fig**). Similarly handled L4 worms on non-auxin NGM plates served as a control. First, we observed that *asna-1*::*mNG*::*AID*;*ieSi57* mothers did not display any morphological defects after 48h exposure to auxin but did produce fewer progeny in comparison to non-treated control animals. On auxin plates all their progeny arrested at the L1 stage (**Fig 2A**). The auxin-mediated depletion of ASNA-1 was effective because the mNeonGreen signal was not visible in any of the previously characterized somatic tissues (**Figs 1 and 2A**). However, we saw a strong mNeonGreen signal in the Z2 and Z3 germline progenitor blast cells (**Fig 2A**), which give rise to the entire germline during development [32]. None of the arrested larvae, even 72 hours after removal of the mothers, showed more than the two cells (Z2 and Z3) in the gonad primordium. The cells could be identified as Z2 and Z3 because they lay in the center of the gonad primordium, were in contact with each other [32], and expressed ASNA-1::mNG::AID while it was depleted in all surrounding somatic cells. To further characterize the effects of ASNA-1 depletion, we sought to establish if *asna-1*::*mNG*::*AID*;*ieSi57* worms exposed to auxin only from the 1st larval stage onwards would display further development and possibly reach later larval stages. Staged L1 hermaphrodites were exposed to 1mM auxin (AUX) for 24h, 48h, and 72h and analyzed at these time points. (**S3B Fig**). Similarly staged unexposed *asna-1*::*mNG*::*AID*;*ieSi57* larvae served as controls. We observed that all animals on AUX plates were still arrested at the L1 stage (**Figs 2B and S4**), whereas control animals reached adulthood. As expected, auxin-treated animals were characterized by a lack of ASNA-1::mNG::AID expression in the somatic tissues and we observed a strong ASNA-1::mNG::AID signal in the germline progenitors (**Fig 3C**). However, in contrast to worms hatched on AUX-containing plates, ASNA-1::mNG::AID was seen in many more germline cells with a maximum of 14 positive cells (**Fig 3C**). In total, 13/20 worms analyzed had more than 4 germline cells. This was in contrast to the mNeonGreen signal only being observed in the two primordial germ cells Z2 and Z3 in animals hatched on AUX plates (**Fig 2A**). However, the larvae in the second experiment did not have the alae that are characteristic of L2 animals indicating that depleting somatic ASNA-1::mNG::AID after the birth of L1 larvae allowed for further development of the germline but did not allow the larvae to reach the L2 larval stage. The size of the gonad was also characteristic of L1 stage larvae. By contrast, auxin-mediated depletion of ASNA-1 from the L4 stage onwards did not block progression to the adult stage.

## Intestine-specific *asna-1* knockdown leads to developmental delay

Our previous research has established that *asna-1* is required cell non-autonomously for growth. Both neuron and intestine-specific promotors driving *asna-1* expression rescue the *asna-1(ok938)* body size phenotype [25], while ASNA-1 is required in the intestine but not the neurons for the rescue of the cisplatin sensitivity phenotype [27]. There is also a requirement for ASNA-1 in the gut for proper tail-anchored protein targeting [28]. Therefore, we sought to establish a role of the intestinal function of ASNA-1 in growth and development to ask whether the L1 arrest phenotype could be ascribed to the TAP targeting function in the intestine. For that purpose, we constructed a strain expressing an intestine-specific TIR1 driver

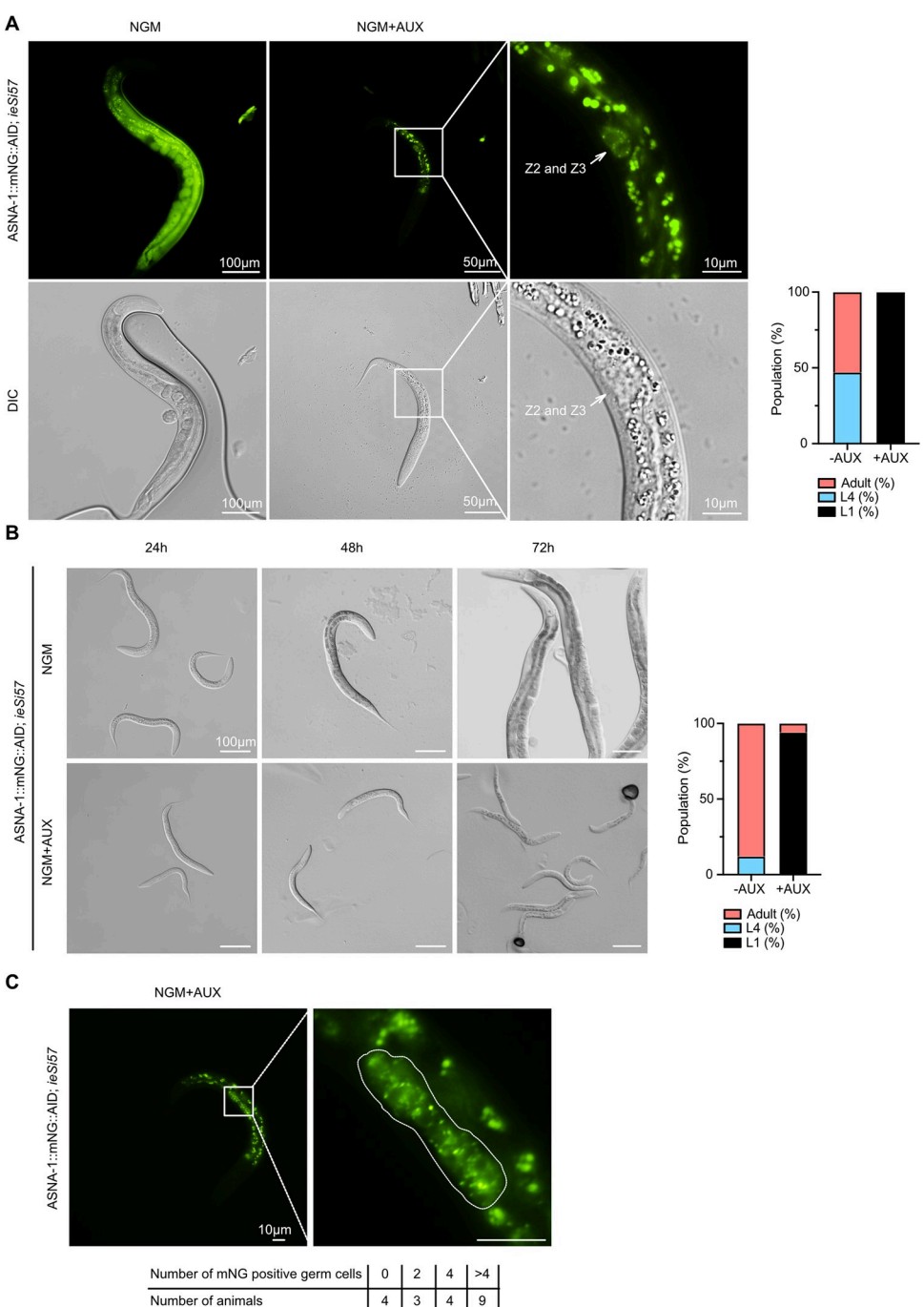

**Fig 2. Soma-specific ASNA-1 depletion leads to an L1 larval arrest.** (A) Representative fluorescent and differential interference contrast (DIC) images of worms expressing ASNA-1::mNeonGreen::AID (*syb2249*) and pan-somatic TIR1 driver *ieSi57* grown on plates without (NGM) or with (NGM+AUX) 1mM auxin. White arrows indicate the Z2 and Z3 cell positions. All auxin-exposed progeny (n = 109) displayed the L1 arrest phenotype. The graph represents the quantification of *syb2249*; *ieSi57* animals population grown on plates without (NGM) or with (NGM+AUX) 1mM auxin. (B) Schematic representation of second experimental setup for auxin-inducible pan-somatic ASNA-1 knockdown. Representative DIC images of worms expressing *syb2249*; *ieSi57* grown on plates without (NGM) or with (NGM+AUX) 1mM auxin at specific time points. The graph represents 72h time point quantification of *syb2249*; *ieSi57* animals population grown on plates without (NGM) or with (NGM+AUX) 1mM auxin. (C) Representative fluorescent images of *syb2249*; *ieSi57* worms grown on plates with 1mM auxin. The dotted line outlines ASNA-1:: mNG::AID positive cells. The table represents the number of animals with a count of ASNA-1::mNG::AID-positive germ cells.

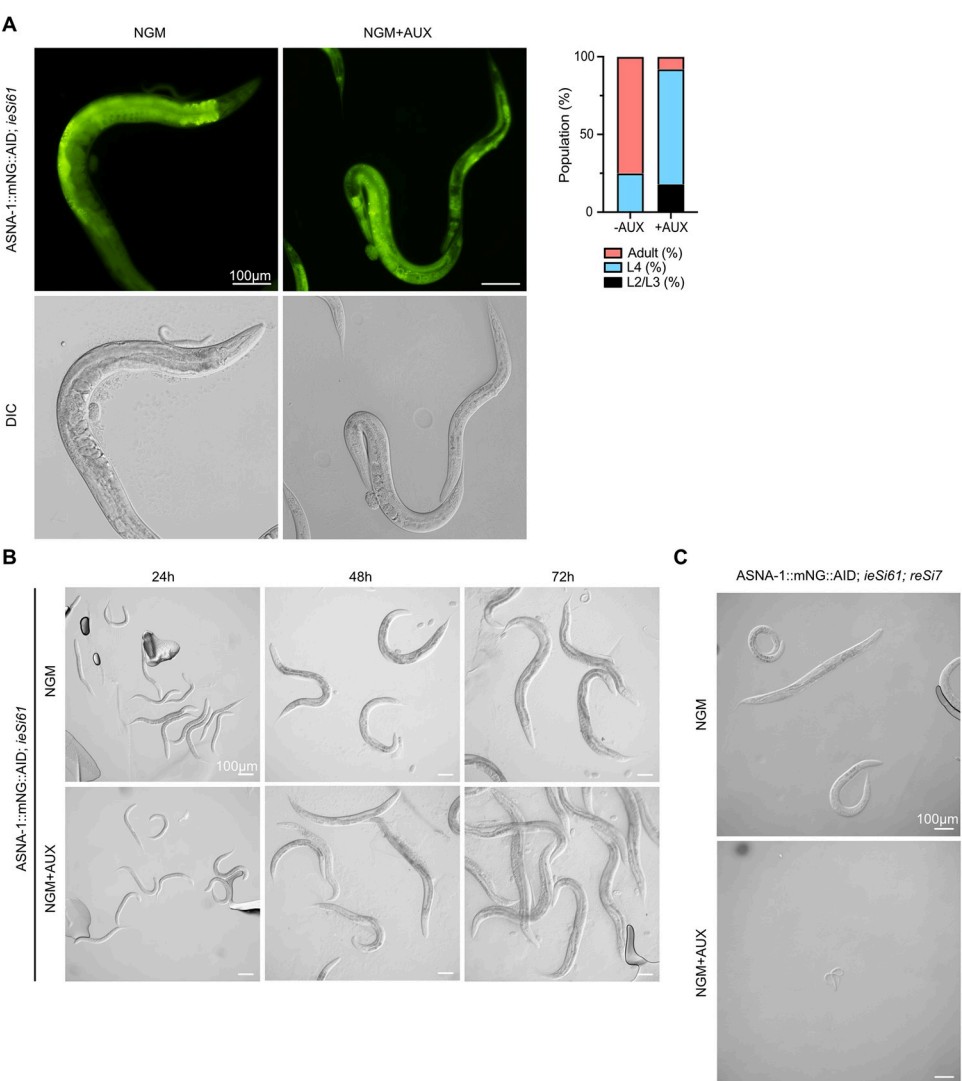

**Fig 3. Intestine-specific ASNA-1 depletion leads to a developmental delay.** (A) Representative fluorescent and differential interference contrast (DIC) images of worms expressing ASNA-1::mNeonGreen::AID (*syb2249*) and TIR1 pan-somatic driver *ieSi61* grown on plates without (NGM) or with (NGM+AUX) 1mM auxin. The graph represents the quantification of *syb2249*; *ieSi61* animals population grown on plates without (NGM) or with (NGM+AUX) 1mM auxin. (B) Schematic representation of second experimental setup for auxin-inducible intestinal ASNA-1 knockdown. Representative DIC images of worms expressing *syb2249*; *ieSi61* grown on plates without (NGM) or with (NGM +AUX) 1mM auxin at specific time points. (C) Representative images of auxin-inducible simultaneous intestinal and neuronal ASNA-1 knockdown. Worms were grown on plates without (NGM) or with (NGM+AUX) 4mM auxin. Auxin-exposed progeny (n>250) displayed the L1 arrest phenotype.

*ieSi61* in the ASNA-1::mNG::AID (*syb2249*) background. To compare pan-somatic depletion of ASNA-1 with gut-specific depletion, as before we exposed *asna-1::mNG::AID;ieSi61* L4 hermaphrodites to 1mM auxin (AUX) for 48h, removed them from the plate and analyzed their progeny 48h after the removal of the mothers (**S3A Fig**). Worms on NGM plates not containing auxin served as a control. Microscopy revealed an absence of mNeonGreen fluorescence in the intestine while pharyngeal and germline expression was visible at normal levels (**Fig 3A**). Intestine-specific ASNA-1 depletion starting in developing embryos led to a delay in development in comparison to control animals (**Fig 3A**). The larvae hatched on these plates took an

additional day to reach adulthood. When L1 larvae were exposed to 1mM AUX (**S3B Fig**), no delay in growth was seen (**Fig 3B**). Thus, there was a requirement for ASNA-1 in the intestine for growth but the requirement was not as stringent as that seen with pan-somatic depletion. Based on that we concluded that growth and development dependent on ASNA-1 was a multi-tissue requirement or that auxin-mediated depletion was not complete.

## Simultaneous depletion of ASNA-1 in the intestine and neurons is equivalent to pan-somatic depletion

Having shown that intestinal depletion of ASNA-1 was not sufficient to produce the L1 growth arrest phenotype, we wished to determine whether we could narrow down the tissue requirements that lead to a strong L1 arrest phenotype seen in *asna-1* mutants [25] upon pan-somatic depletion. To do this we first depleted by auxin treatment ASNA-1::mNG::AID in neurons using the pan-neuronal driver *reSi7* [31]. *asna-1::mNG::AID (syb2249); reSi7* worms were exposed to auxin from the time of hatching by putting 4th larval stage mothers on auxin-supplemented NGM plates. The worms produced a normal number of progeny and the progeny displayed no growth defects or developmental delays. To ask whether the 1st larval stage arrest was caused by simultaneously depleting ASNA-1 in both the intestine and neurons, we next made and tested a strain of the genotype *asna-1::mNG::AID (syb2249);reSi7;ieSi61*. In this strain, the two TIR1 drivers promoted the simultaneous depletion of ASNA-1 in both tissues. In these worms, we observed that when 4th larval stage hermaphrodites were placed on auxin-supplemented NGM plates (**S3A Fig**), they still produced a large number of progeny as adults but all the progeny (n>250) arrested as 1st larval stage sized animals. This arrest reproduced the effect of depleting ASNA-1 in all somatic tissues (**Fig 3C**) and indicated that both a neuropeptide secretion defect (such as DAF-28 secretion) and a defect in the trafficking of SEC-61β alone or along with other TAPs is required for the control of growth at the 1st larval stage. To obtain support for this notion, we next examined an *asna-1* mutant with a known defect in TAP insertion and determined its insulin/IGF signaling status.

## Insulin signaling is normal and germline development defects are less severe in *asna-1(ΔHis164)* compared to *asna-1* deletion mutants

Our previous work shows that *asna-1(ΔHis164)* mutants are as sensitive to cisplatin as deletion mutants in the gene and have a severe TAP insertion defect [28]. Further, we find that the ASNA-1$^{\Delta His164}$::GFP protein largely exists in the oxidized state [28]. Transgene expressing *asna-1$^{\Delta His164}$*::GFP rescues the *asna-1* growth phenotype of the deletion mutant indicating indirectly that the point mutant protein can restore insulin signaling [27]. To examine the growth phenotypes of *asna-1(ΔHis164)* mutants, we directedly measured the insulin secretion and signaling efficiency by analysis of DAF-16::GFP localization and DAF-28/insulin::GFP secretion. The cytoplasmic localization of DAF-16::GFP indicates high IIS levels since DAF-16::GFP is localized to nuclei in mutants with insulin signaling defects [33]. The uptake of secreted DAF-28/insulin::GFP by coelomocytes, which are found in the open circulatory system, reports on insulin secretion [25]. In *asna-1(ΔHis164)* animals DAF-16::GFP protein was cytoplasmic and the DAF-28/insulin::GFP protein was secreted and taken up by coelomocytes at normal levels (**Figs 4A, 4B, S5A, and S5B**). The analysis of both reporters showed that insulin secretion and signaling were unaffected in *asna-1(ΔHis164)* mutants in comparison to the strong defects in *asna-1* deletion mutants. Although both mutants did differ in IIS status, both the *asna-1(ΔHis164)* and *asna-1* deletion mutants were sterile. The bigger size of the *asna-1 (ΔHis164)* worms (**Fig 4C**) suggested that germline defects in the two mutants differ significantly. We sought to understand these differences by examining the germline phenotype of

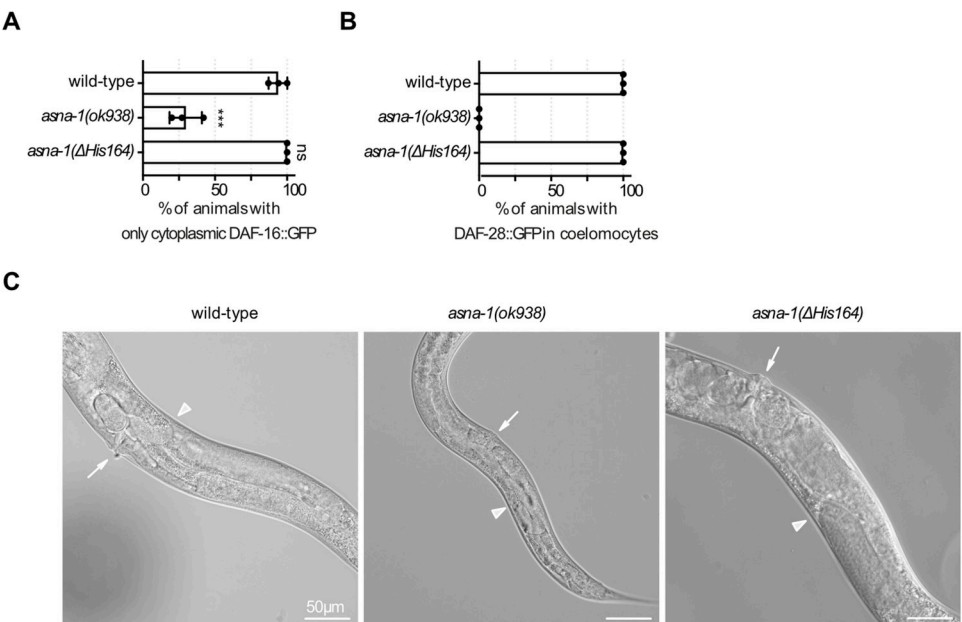

**Fig 4. Comparison of germline and insulin/IIS defects *in asna-1(Δ His164)* and *asna-1(ok938)* deletion mutants.** Percentage of 1-day-old adults with indicated genotypes displaying (A) only cytoplasmic localized DAF-16::GFP and (B) percentage of 1-day-old adults with secreted DAF-28::GFP in coelomocytes. Experiments were performed in triplicate. Bars represent mean ± SD. (A) Statistical significance was determined by the one-way ANOVA followed by Bonferroni post-hoc correction. (C) Representative DIC pictures of adults showing the body size of wild type, *asna-1 (ok938)*, and *asna-1(ΔHis164)* strains.

*asna-1(ok938)* and *asna-1(ΔHis164)* animals. For this purpose, we used the transgenes *ltIs37* and *ltIs38* to simultaneously monitor germ cell membranes and germline nuclei *in vivo*. *asna-1 (ok938)* animals did not produce any oocytes or sperm and showed a gonad migration defect (**Fig 4C**). *asna-1(ΔHis164)* also showed a gonad migration defect but the germlines contained oocytes and sperm as well as non-fertilized embryo-type structures in the uterus. These cells contained only one nucleus and displayed no embryonic divisions (**S6A Fig**). Ovulation, the transit of the ovum through the spermatheca, and entry of oocytes into the uterus require the presence of sperm [34]. This led us to investigate whether *asna-1(ΔHis164)* mutants might be defective in fertilization, taking into consideration the strong ASNA-1 expression in mature oocytes, sperm, and the spermatheca in wild-type worms (**Figs 1 and S2**). Indeed, the *ltIs37* marker expressing *pie-1p::mCherry::his-58* to visualize germline nuclei showed that sperm was present in the spermatheca in *asna-1(ΔHis164)* animals as in *asna-1(+)* (**S6B Fig**). However, since the oocytes that passed through the spermatheca did not contain more than one nucleus or show any embryonic divisions, we concluded that there was a defect either in the oocytes or sperm of *asna-1(ΔHis164)* animals. The difference in germline defect between point mutants and deletion mutants stands in striking contrast to the similarity in cisplatin sensitivity phenotype of these two mutants [28] indicating that different functions can be independently targeted.

## Analysis of ASNA-1 point mutants reveals the importance of a conserved alanine for cisplatin resistance

We have previously shown that *C. elegans* ASNA-1 exists in two redox states: reduced (ASNA-1$^{RED}$) and oxidized (ASNA-1$^{OX}$), and that important biological functions of the protein are

separable based on the redox state [28]. As mentioned, these conclusions were drawn based on an analysis of the *asna-1* single point mutant: *asna-1(ΔHis164)*, which also showed the sterility phenotype described above. Therefore, we wondered if this was a unique characteristic of mutating His164 or whether any other single point mutation in the ASNA-1 protein would also be able to completely separate the important biological functions of ASNA-1: insulin secretion, TAP targeting, and cisplatin sensitivity while not causing sterility in the worm. For this reason, we examined a set of seven strains generated by the Million Mutation Project, each carrying a missense point mutation in ASNA-1 along with other mutations in the genome [35] (**S1 Table**). Of these seven, only in four mutants were the mutated amino acids conserved with human ASNA1 (**S1 Table** and **S7A Fig**). None of the single amino acid changes caused reduced steady-state levels of the ASNA-1 protein (**S8 Fig**). We performed a cisplatin survival analysis of all seven strains to ask if any of those mutants could increase sensitivity to cisplatin treatment. The $LD_{50}$ of cisplatin for *asna-1(ok938)* deletion mutants is 300 μg/mL after 24h treatment [27] and hence we used this concentration for our experiments. We found that only two strains, *asna-1(gk687101)* and *asna-1(gk592672)*, showed increased cisplatin sensitivity after 24h exposure to the drug (**Fig 5A**). Since *asna-1(gk592672)* strain produced a more severe sensitivity phenotype, we focused on this point mutation in our further analysis. The *asna-1 (gk592672)* mutation is an alanine to valine change, at a highly conserved position 63 which corresponds to alanine 82 in the human homolog and is positioned close to the Switch I domain (**S7A Fig**). This domain plays an essential role in the activation of ATP hydrolysis during TA protein targeting. We used the GnomAD browser (https://gnomad.broadinstitute.org/ ) to assess the effect of this point mutation in the human homolog of ASNA-1. The browser provides not only a list of variations but also a critical view of the clinical application [36]. In the exome sequence data for 60,706 individuals, ASNA1 has 79 observed single nucleotide variants (SNVs) that resulted in missense mutations. In those, A82 was detected to be mutated to threonine (T) (A82T) with two allele counts in the European (non-Finnish) population. Poy-Phen-2 (http://genetics.bwh.harvard.edu/pph2/) is a tool that predicts the possible impact of amino acid substitution on the structure and function of human proteins. The human ASNA1 (A82T) mutation was classified as 'probably damaging'. This was also true for the alanine to valine substitution. Mindful of other mutations in other genes in the *asna-1(gk592672)* containing strain we outcrossed the strain 13 times and performed the cisplatin sensitivity assay again. Indeed, the outcrossed strain, hereafter called *asna-1(A63V)*, was as sensitive to cisplatin exposure as the *asna-1(ok938)* deletion mutant. The cisplatin sensitivity phenotype was rescued by a single copy of wild-type ASNA-1 fused to GFP (*knuSi184*) expressed from a transgene (**Fig 5B**). This transgene also rescued the ASNA-1 protein null mutant for the cisplatin sensitivity phenotype (**Fig 5B**). We concluded that change in a highly conserved alanine at position 63 was the genetic locus causing cisplatin sensitivity as severe as that seen in the deletion mutant and no other mutation in the genetic background contributed to the phenotype.

## *asna-1(A63V)* mutants have a normal brood size and insulin/IIS signaling and secretion

To obtain more evidence for the separable functions of ASNA-1, we characterized the *asna-1 (A63V)* strain. As in the case of the *asna-1(ΔHis164)* protein [28], the A63V mutation did not affect the steady-state levels of the protein (**Figs 5C and S9**). ASNA-1^A63V::GFP expressed from a multi-copy transgene was detected in the ER (**S7B Fig**) in the same manner as the wild-type protein ASNA-1::GFP. *asna-1(A63V)* animals did not display the enhanced ER stress phenotypes (**S7C** and **S7D Fig**) shown by the *asna-1(ok938)* deletion mutant [28]. While *asna-1 (A63V)* animals were as sensitive to cisplatin as the deletion mutant (**Fig 5B**), they were fertile

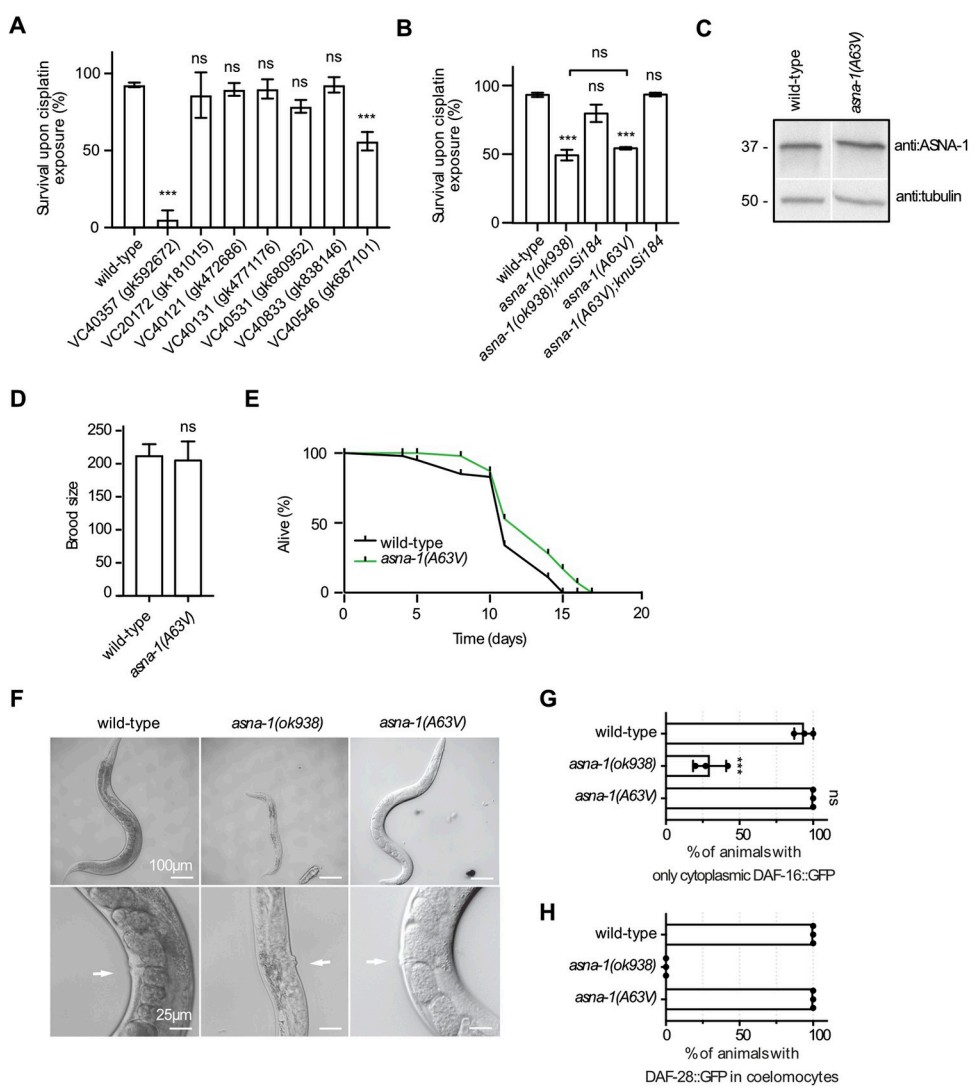

**Fig 5. Analysis of ASNA-1 point mutants reveals the importance of a conserved alanine for cisplatin resistance.**
(A) Analysis of cisplatin sensitivity phenotype of seven million mutation project strains bearing point mutations in
*asna-1*. (B) Rescue of the cisplatin sensitivity phenotype of *asna-1(ok938)* deletion mutants and *asna-1(gk592672*
/A63V) mutants with a single copy ASNA-1(+) transgene *knuSi184*. Bars represent mean survival ± SD of 1-day-old
adult animals exposed to 300 μg/mL of cisplatin for 24h. Statistical significance was determined by the one-way
ANOVA followed by Bonferroni post-hoc correction. Survival experiments were performed in triplicate. (C) Western
blot analysis to estimate ASNA-1 levels in wild-type and *asna-1(A63V)* mutants. The blot was probed with an anti-
ASNA-1 antibody. Tubulin was used as a loading control. For a full uncropped western blot image, see S9 Fig. (D)
Brood size analysis of wild-type and *asna-1(A63V)* mutants (n = 5). Statistical significance was determined by the
independent two-sample t-test. Bars represent mean ± SD. (E) Life span analysis of wild-type (n = 40) and *asna-1
(A63V)* (n = 40) animals. (F) Top panel shows representative DIC pictures of adults to show the body size of wild type,
*asna-1(ok938)*, and *asna-1(A63V)* strains. The magnification of the germline of the animals is shown in the bottom
panel. White arrows indicate the vulva position for orientation purposes. (G-H) Percentage of 1-day-old adults with
indicated genotypes displaying (G) only cytoplasmic localized DAF-16::GFP and (H) percentage of 1-day-old adults
with secreted DAF-28::GFP in coelomocytes. Experiments were performed in triplicate. Bars represent mean ± SD. (G)
Statistical significance was determined by the one-way ANOVA followed by Bonferroni post-hoc correction.

with a normal brood size and life span (**Fig 5D and 5E**), and had properly developed germlines
(**Fig 5F**). This characteristic opened up the possibility of separating the cisplatin, TAP inser-
tion, and insulin functions of ASNA-1 without compromising on the sterility phenotypes seen

in *asna-1(ok938)* or *asna-1(ΔHis164)* mutants [28,37] Importantly, *asna-1(A63V)* mutants had neither the insulin signaling nor the insulin secretion defect (**Figs 5, 5H, S5A, and S5B**), which was measured respectively by cytosolic/nuclear localization of DAF-16::GFP and uptake of DAF-28::GFP by the coelomocytes. This evidence led to the conclusion that indeed functions of ASNA-1 in cisplatin sensitivity and insulin signaling/secretion were completely separable in animals without the compromised function in the germline and further that the cisplatin sensitivity of *asna-1* mutants was purely a soma-specific defect since the germline is normal in *asna-1(A63V)* mutants.

## *asna-1(A63V)* mutants separate the TAP insertion function from insulin secretion

We have previously shown that *asna-1(ok938)* and *asna-1(ΔHis164)* mutants displayed a strong defect in the targeting of the TAP SEC-61β [28]. This conclusion was supported by two independent tests. First, Pearson colocalization analysis following confocal microscopy of worms co-expressing GFP::SEC-61β with the ER marker, mCherry::SP12. Second, N-linked glycosylation analysis of SEC-61β to monitor the insertion of SEC-61β into the ER membrane [28]. Since *asna-1(ok938)* and *asna-1(ΔHis164)* mutants share the cisplatin sensitivity phenotype of *asna-1(A63V)* mutants, we therefore wondered if *asna-1(A63V)* animals could support ASNA-1-dependent SEC-61β insertion. Pearson correlation analysis of confocal microscope images of the two co-expressed transgenes (GFP::SEC-61β and mCherry::SP12) in *asna-1 (A63V)* mutants revealed a significantly defective TAP targeting phenotype (**Fig 6A and 6B**). We next measured the direct insertion of the SEC-61β protein into the ER membrane [28]. We used a transgene (*rawEx64*) expressing a tagged SEC-61β in which the beta-opsin tag is inserted just before the stop codon. The opsin tag contains a reactive arginine which is glycosylated by ER luminal enzymes when the C-terminus is exposed to the ER lumen. This reports on the proper insertion of the SEC-61β protein into the ER membrane. The glycosylation event is revealed by the change in electrophoretic mobility of the glycosylated tagged protein. This analysis revealed that ER insertion of SEC-61β was decreased in the *asna-1(A63V)* mutant background (**S10 Fig**) consistent with the conclusions drawn from confocal microscopy analysis.

## The A63V mutation alters the subcellular localization of ASNA-1

ASNA-1 physically interacts with the ER membrane resident receptor WRB-1 [28]. We next investigated whether the defect in GFP::SEC-61β targeting might be also associated with the reduced association of the ASNA-1(A63V) protein with the membrane. Separation of cytoplasmic from membrane fractions by ultracentrifugation revealed significantly reduced membrane association of the ASNA-1$^{A63V}$::GFP protein compared to that seen in ASNA-1$^{+}$::GFP (**Figs 6C and S11**). We concluded that *asna-1(A63V)* mutants shared the TAP insertion defect seen in the *asna-1* deletion mutants likely via a reduced ability to associate with ER membranes. This defect was manifested without affecting the insulin signaling and indeed those two vital functions of ASNA-1 were likely independent of each other.

## The ASNA-1(A63V) mutant protein preferentially exists in the oxidized state

We have previously shown that *C. elegans* ASNA-1 is a redox-modulated protein and its functions are separable based on the redox state of the protein [28]. We obtained evidence for this functional separation based on the analysis of *asna-1(ΔHis164)* point mutants in which the

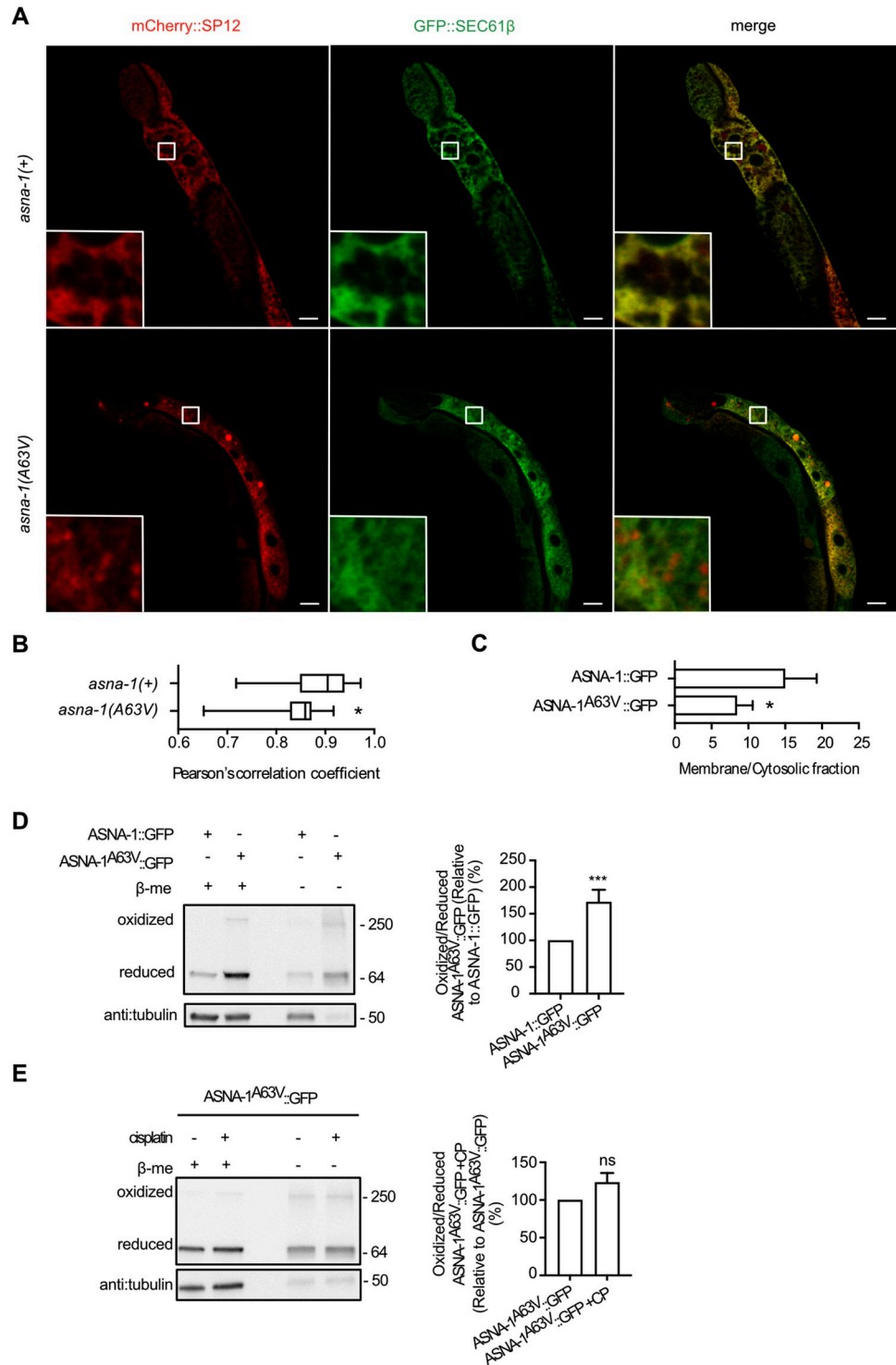

**Fig 6. *asna-1(A63V)* mutants are defective for TAP insertion but have normal insulin secretion.** (A) Representative confocal images of *asna-1(+)*, and *asna-1(A63V)* 1-day-old adult animals co-expressing GFP::SEC-61β and mCherry::SP12 in the intestine. Imaging was performed in int8 and int9 cells. Scale: 10 μm and 60 μm for magnification. (B) Pearson's correlation analysis of GFP::SEC-61β and mCherry::SP12 co-localization in *asna-1(+)* (n = 13) and *asna-1(A63V)* (n = 19). The box plot represents the average Pearson correlation coefficient (R) of the indicated strains. Statistical significance was determined by the Mann-Whitney test. (C) Band intensity quantification of membrane/cytosolic fraction of ASNA-1::GFP and ASNA-1<sup>A63V</sup>::GFP based on the western blots presented in <u>S11 Fig</u>. Statistical significance was determined by the independent two-sample t-test. Error bars represent ± SD. (D)

Representative western blot after reducing and non-reducing SDS-PAGE to detect levels of oxidized and reduced ASNA-1$^{A63V}$::GFP. Control worms expressed the ASNA-1(+)::GFP transgene. Blots were probed with anti:GFP antibody and tubulin was the loading control. For full uncropped blot source images including replicates used for quantification, see S11 Fig. Graph represents the band intensity quantification of oxidized/reduced ASNA-1$^{A63V}$::GFP. Statistical significance was determined by the independent two-sample t-test. Experiments were performed in triplicate. Bars represent ± SD. (E) Representative western blot after reducing and non-reducing SDS-PAGE to detect levels of oxidized and reduced ASNA-1$^{A63V}$::GFP upon cisplatin (+CP) treatment (300 μg/mL for 6h). Control worms expressed the ASNA-1$^{A63V}$::GFP transgene. Blots were probed with anti:GFP antibody and tubulin was the loading control. For full uncropped blot source images including replicates used for quantification, see S13 Fig. Graph represents the band intensity quantification of oxidized/reduced ASNA-1$^{A63V}$::GFP. Statistical significance was determined by the independent two-sample t-test. Experiments were performed in triplicate. Bars represent ± SD.

protein preferentially exists in an oxidized state. This mutant was defective for TAP insertion and cisplatin resistance [28] and as shown in Fig 4, *ΔHis164* mutants were normal for insulin secretion/signaling function. We therefore wondered, based on the analysis of *asna-1(A63V)* mutants, if in *asna-1(A63V)* animals the separation of ASNA-1 functions is reflected in changes in the oxidative state of the protein. To address this, we performed non-reducing SDS-PAGE to evaluate the oxidative state of the ASNA-1$^{A63V}$::GFP protein. Indeed, the analysis revealed that more of the ASNA-1$^{A63V}$ protein exists in the oxidized state (**Figs 6D and S11**). Strikingly, the shift in ASNA-1$^{A63V}$::GFP to the oxidized state occurred in the absence of an increased expression of markers of ER stress (*hsp-4*) (**S7C and S7D Fig**), or mitochondrial stress markers (*hsp-6* and *hsp-60*) (**S7E Fig**), whereas oxidative stress markers (*gst-4*, *gst-30*, and *gst-38*) were slightly increased (**S7F Fig**). Cisplatin treatment was not able to increase even further the amount of oxidized ASNA-1$^{A63V}$::GFP (**Fig 6E**). We concluded that the A63V mutants in ASNA-1 had inherently high ASNA-1$^{OX}$ levels, which resulted in sensitivity to cisplatin. They also had a TAP insertion defect but maintained not only normal insulin secretion function but also did not have a compromised germline phenotype. This conclusion is consistent with our previous analysis which showed that lower levels of reduced ASNA-1 had no impact on growth and DAF-28 secretion.

### *in silico* modeling of ASNA-1(A63V) and ASNA-1(ΔHis164) proteins predicts the importance of Switch I and Switch II domains in function separation

The model of the wild-type ASNA-1 protein assembled as a dimer shows that the dimerization interface presents three main interaction regions: the zinc atom coordination, the ATP binding site, and the Histidine164 contacts (**S12 Fig**). The model obtained by the single point mutation A63V showed that the substitution of alanine by valine reduced the distance between the helix α3 and the helix α2 by the addition of the isopropyl group at the position 63 (helix α3) pointing directly towards the Thr32 (helix α2) in the P-loop motif (**Fig 7A**). The model proposed a movement correction of the threonine hydroxyl group induced by this close contact with the Valine. The Thr32 plays a key role in the coordination of the Mg atom (**Figs 7A and S12**), its hydration state, and thereby the binding and catalysis of the ATP. The binding of ATP is a key event in the formation of the closed dimer and the formation of the closed interface by its direct participation in it. We conclude that A63V change might influence the P-loop structure and affect ATP binding.

The wild-type model showed the close interaction between the His164 and His164' of each monomer to perform the dimer interface (**Fig 7B**). The model of ASNA-1 presenting the deletion of the His164 showed that the loss of histidine implies a loss of one of the key amino acids that create the dimeric interface and does not suggest the formation of compensatory

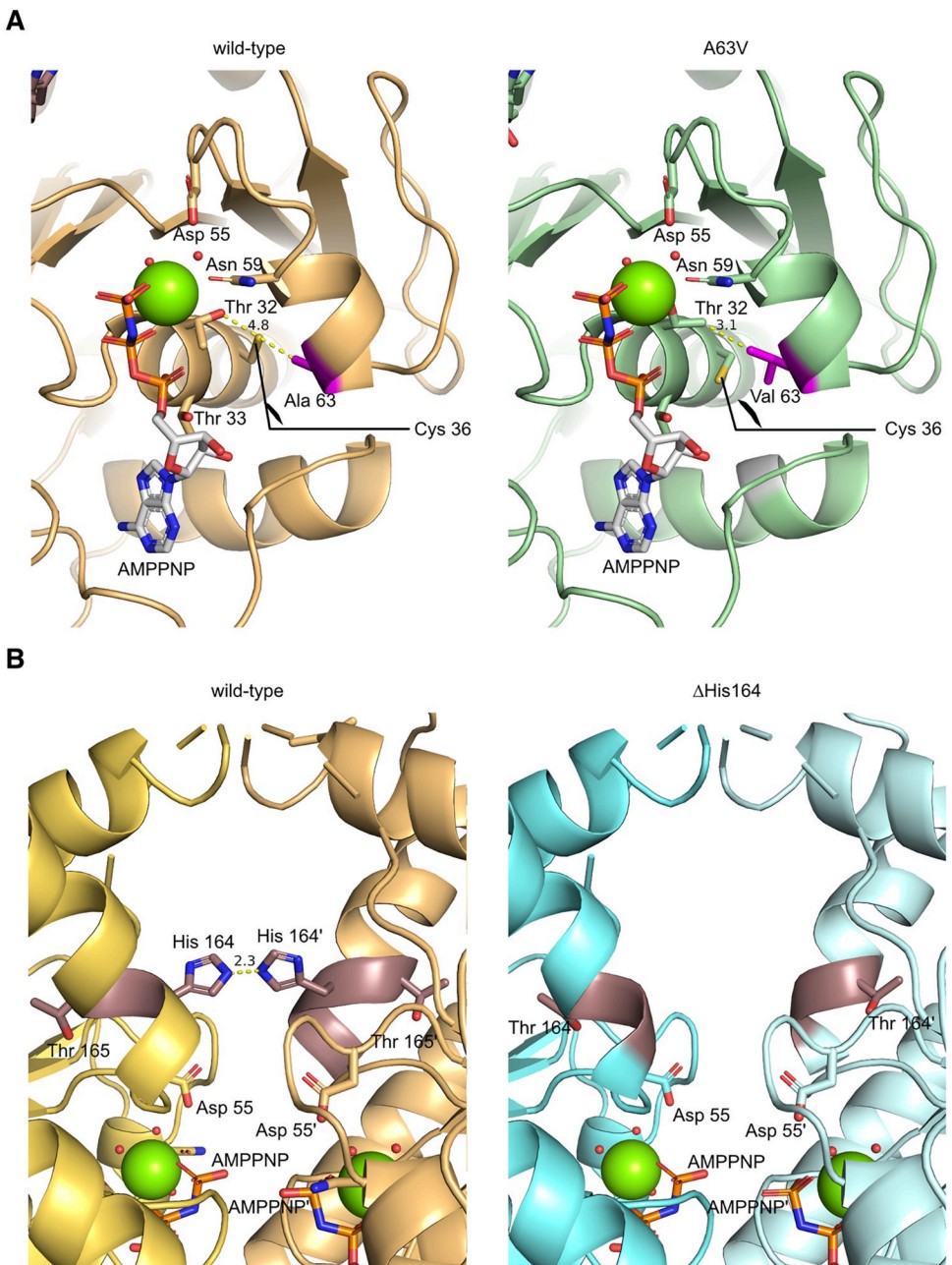

**Fig 7. Structural representation of the effect of the A63V and ΔH164 mutations in the ATP-bound ASNA-1 dimer complex.** The *in-silico* models of *C. elegans* ASNA-1 mutants show the structural impact of these mutations in the key processes of ASNA-1 ATPase activity and the dimer complex formation. (A) In the left panel, the model of an ASNA-1 wild-type protein monomer in complex with Mg (green) and AMPPNP (white) is shown in light orange. The amino acid Ala63 is shown in pink. The amino acids depicted show the influence of the amino acid at position 63 on the amino acids coordinating the Mg atom. In the right panel, the monomer of ASNA-1(A63V) is represented in green and the Val63 in pink. The distances between the amino acid at position 63 and Thr32 are shown in yellow. (B) In the left panel each monomer of the ASNA-1 wild-type dimer in complex with Mg (green) and AMPPNP in light orange and yellow-orange is represented by, in deep salmon is shown the position of His164. The distance between the His164 of each monomer is displayed in yellow. In the right panel, each monomer of ASNA-1(ΔHis164) is colored in cyan and light cyan, and the amino acid Thr164 is colored in dark salmon.

interactions. (**Fig 7B**). Therefore, the deletion of this amino acid would affect the stability of the closed conformation and probably its formation in a negative fashion.

## Discussion

An intensively studied role of ASNA-1 homologs is in the targeting of a subset of tail-anchored proteins to the ER membrane and beyond. However, it has also been clear that ASNA-1 and its homologs have roles that are not connected to its tail-anchored targeting function. These include modulation of chloride channel function, a holdase function for aggregated proteins, and the binding to the ALS-related protein VAPB protein among others [12,21,38]. Earlier work from our lab has also addressed this issue and separated insulin secretion from cisplatin response [28]. Since ASNA-1 promotes the ER targeting of SEC-61β in the intestine [28] and is also required for the neuronal secretion of DAF-28/insulin [25], one objective of this study was to examine further the relative contributions of different tissues for growth and reproduction. A second major goal was to discover whether single point mutations in *C. elegans* ASNA-1 will fulfill the requirements for ASNA-1 function for cisplatin response separate from its roles in growth and reproduction.

First, we characterized the expression pattern of ASNA-1 in somatic tissues as well as in the germline. We showed that general somatic tissue depletion led to the strict L1 arrest phenotype identical to that seen in *asna-1* mutants depleted from both maternal and zygotic contributions. We also found that intestine-specific depletion of ASNA-1 led to a growth defect but that it was not as severe as pan-somatic depletion, raising the possibility that the contribution of ASNA-1 to the L1 arrest likely came from both neurons and the intestine. This was further confirmed by the simultaneous depletion of ASNA-1 in the intestine and neurons. Pan-somatic depletion after embryogenesis showed that *asna-1* was required for the development of worms from the L1 to the L2 stage but not from the L4 larval stage to adulthood. This indicated that ASNA-1 cannot be regarded as a housekeeping gene whose function is needed in a constitutive manner.

The million mutation project (MMP) is a powerful resource for exploring mutations in almost every *C. elegans* gene [35]. It led to a collection of a large number of mutations in over 2000 whole genome sequenced strains. We tested seven single-point mutants in ASNA-1 generated by this project for cisplatin sensitivity (S1 Table). Since all these seven strains were homozygous for the point mutants and were fertile, it was very likely that none of the seven point mutants affected reproduction. Given that the mutants were fertile we expected only mild changes in the overall effect on the function of the mutant ASNA-1. However, we cannot rule out the possibility that a compensatory second site mutant might act in some cases. Only two of the seven strains were cisplatin sensitive. We concluded that heavy mutation load *per se*, which is present in all the strains will not result in cisplatin sensitivity, and worms that were unhealthy were not inherently prone to cisplatin sensitivity due to some non-specific reason. In the screen of viable ASNA-1 mutants, which would be able to separate the clinically relevant functions of ASNA-1 in insulin secretion and cisplatin resistance, we identified *asna-1(A63V)* as important in function separation.

The *asna-1(A63V)* mutant produces a protein that preferentially exists in the oxidized state, has a strong cisplatin sensitivity phenotype and tail-anchored protein insertion defect, and has a defective subcellular localization. Importantly, despite these defects, the mutation did not affect IIS activity nor caused any growth or developmental problems. This allowed us to genetically separate *asna-1* functions even further without compromising the viability of the worms. Evidence for functional separation emerged from the earlier study of another point mutant: *asna-1(ΔHis164)*. We had previously shown that this mutant too is cisplatin sensitive and the

**Table 1. Summary of phenotypes of *asna-1* deletion and point mutants discussed in the paper to highlight the different phenotypes displayed with differing levels of ASNA-1 activity.** Cisplatin sensitivity is independent of germline development ability and insulin secretion ability but linked to the TAP targeting phenotype. Animals with a large increase in the ratio of oxidized v/s reduced ASNA-1 will still display the cisplatin sensitivity phenotype indicating that increased ASNA-1 levels do not protect these mutants from cisplatin induced death.

|  | cisplatin sensitivity | TAP targeting | insulin secretion | germline development | oxidized/reduced ASNA-1 ratio |
|---|---|---|---|---|---|
| wild-type | - | + | + | +++ | + |
| *asna-1(ok938)* | + | - | - | + | NR |
| *asna-1(ΔHis164)* | + | - | + | ++ | +++ |
| *asna-1(A63V)* | + | - | + | +++ | ++ |

NR–not relevant

mutant protein preferentially accumulates in the oxidized state [28]. Here we present evidence that the IIS functions of ASNA-1 remain largely intact in the *ΔHis164* mutant and the germline development is substantially better than that seen in the *asna-1* deletion mutant since oocyte and sperm are formed and ovulation (the exit of oocytes from the ovary) also occurs. Taken together our work shows that a marked shift towards the accumulation of higher levels of oxidized ASNA-1 does not perturb insulin secretion and growth but does have an impact on TAP targeting and the cisplatin sensitivity phenotype (Table 1 and S14 Fig).

*In silico* modeling showed that the A63V mutation, which is close to the Switch I motif, affects the coordination of a conserved Threonine in the P-loop motif with possible effects on ATP binding and hydrolysis. Further, another function separating mutation in highly conserved Histidine 164 lies in the Switch II motif and had no effect on insulin secretion and signaling. This indicates that Switch I and Switch II and P-loop motifs are excellent druggable targets that increase cisplatin sensitivity while sparing the insulin secretion role of ASNA-1. Indeed, ATPases have been an attractive target for drug discovery for many years considering that their dysregulation is at the origin of many human diseases. Already several ATPase inhibitors have been described [39–42] and have been shown to be beneficial in the treatment of various diseases. One of the possibilities to target the Switch I/II domains in ASNA1 in order to separate the ASNA1 functions might be the use of small molecule, competitive ATP inhibitors. In particular, it would be beneficial for the treatment of cancer types that are resistant to chemotherapy. One of the alternatives would be a synthesis of a specific ATP inhibitor produg, specific for Switch I and Switch II domains present in ASNA1, which would also contain a small peptide carrier. The activation of the drug would require the cleavage of the peptide by specific proteases which would allow for the design of tissue-specific ATP inhibitor. Full validation of this strategy is still needed, but it seems clear that targeting the domains important for ATPase activity in ASNA1 is an appealing approach.

To be able to develop targeted cancer therapies, there is a need to have a better understanding of which cellular pathways are involved in the cisplatin response and drug resistance. Based on the results presented here we believe that separating ASNA1 functions in cisplatin response in insulin secretion by targeting the ATPase domain of the protein by small molecule drugs that act as competitive ATP inhibitors will allow us to re-sensitize tumors to the cisplatin treatment leaving insulin secretion undisturbed.

## Materials and methods

### *C. elegans* strains maintenance and synchronization

The Bristol strain (*N2*) was the wild-type. The *asna-1(gk592672)* containing strain VC40357 was outcrossed 13 times using *unc-32(e189)* and *oxTi719* as balancers and kept *in trans* to the

*hT2(qIs48)* balancer. ASNA-1$^{A63V}$::GFP was generated by mutagenesis of pVB222GK [25] to introduce the A63V change using the QuickChange II Kit (Agilent Technologies). The resulting plasmid (pGK200) was used to generate the extrachromosomal array *rawEx8*. Genomic integration of *rawEx8* yielded *rawIs16*. *rawIs16* worms were outcrossed 4 times before analysis. The single-copy *asna-1::gfp (knuSi184)* contained 1.4 kb upstream promoter sequence driving the genomic *asna-1* coding region fused to GFP just before the stop codon followed by the *tbb-2* 3' UTR. This construct was inserted on chromosome II at the *ttTi5605* locus using MosSCI technology [43] by Knudra Transgenics. *rawEx64* expressed *vha-6p-3xFlag::SEC-61β::opsin* as an extrachromosomal array and was outcrossed 2 times before use [28]. All *C. elegans* strains used in this study are listed in **S2 Table**. Worms were cultured under standard conditions at 20˚C on nematode growth media (NGM) plates unless stated otherwise and the *E. coli* strain OP50 was used as a food source. *Synchronization*: Synchronous larval populations were obtained by gravity separation as described previously [44].

## TAP-targeting analysis

The assay was performed as described previously [28]. In short, live 1-day-old adult animals were sedated in 1 mM Levamisole/M9 and mounted onto 2% agarose pads. The int8 and int9 cells in the posterior intestine were imaged. The fluorescence signals were analyzed at 488 nm and 555 nm by the LSM700 Confocal Laser Scanning Microscope (Carl Zeiss) with LD C-Apochromat 40x/1.1 W Corr. objective. Image processing of Z-stacks was performed with the ZEN Lite program (Zeiss). Correlation quantification was done using Volocity software (PerkinElmer). Correlation quantification was done using the Automatic Thresholding [45] method to set thresholds objectively.

## Insulin assays

Worms harboring integrated *daf-16*::*gfp (zIs356)* and *daf-28*::*gfp (svIs69)* arrays were grown at 20˚C and imaged using a Nikon microscope, equipped with Hammamatsu Orca flash 4.0 camera. *daf-16::gfp* animals were analyzed within 10 minutes after mounting to avoid artifacts due to stress. DAF-28::GFP uptake by coelomocytes was scored in adult worms as described previously [25].

## Glycosylation analysis of tagged SEC61β

Young adult worms expressing *3xFlag*::*SEC-61β*::*opsin* from the *rawEx64* transgene were homogenized and taken for protein concentration determination. All samples were solubilized in Laemmli buffer, separated by SDS-PAGE followed by western transfer and detection by immunoblotting with anti-Flag (F1804, Sigma) antibody.

## Western blot analysis

*Reducing SDS-PAGE*: Synchronized young adult worms were homogenized and protein concentration was determined using the BCA assay (Thermo Scientific). Samples were boiled for 10 min in reducing loading buffer (SDS/ β-mercaptoethanol). Proteins were separated by SDS-PAGE and blotted onto PVDF membranes. *Non-reducing SDS-PAGE*: Lysates were boiled for 10 min in non-reducing (without β-mercaptoethanol) loading buffer and cooled to room temperature for 10 min. To protect free cysteine thiols from post-lysis oxidation, iodoacetamide was added to the samples at a final concentration of 25 mM followed by a 30 min incubation in darkness at room temperature. Proteins were separated by SDS-PAGE and blotted onto nitrocellulose membranes. *Antibodies*: anti-ASNA-1 antibody [25], anti-GFP antibody

(3H9, Chromotek), and anti-Flag M2 antibody (F1804, Sigma) were used. To assess equal loading, membranes were stripped and probed with anti-alpha tubulin (T5168, Sigma). Band quantification was performed using ImageJ software [46].

## Subcellular fractionation

Young adult animals grown at 20˚C were harvested, washed, and lysed in extraction buffer (50 mM Tris, pH7.2, 250 mM sucrose, 2 mM EDTA). Supernatants were centrifuged for 60 min at 100,000xg at 4˚C. The supernatant fraction was concentrated using Vivaspin Concentrators (Sigma). The pellet fraction was resuspended in 1x Laemmli buffer (Biorad). Proteins in both fractions were separated by SDS-PAGE and blotted onto PVDF membranes. Proteins were detected using an anti-GFP antibody (3H9, Chromotek) or anti-alpha tubulin (T5168, Sigma).

## RNA isolation and quantitative PCR

Total RNA was extracted using Aurum Total RNA Mini Kit (BioRad). cDNA was synthesized using the iScript cDNA Synthesis Kit (BioRad). qPCR was performed on a CFX Connect machine (BioRad) instrument using KAPA SYBR FAST qPCR Kit (KapaBiosystems) with the comparative Ct method and normalization to the housekeeping gene *F44B9.5*. All samples were tested in triplicates.

## Auxin treatment

Animals were transferred to NGM plates containing indicated concentration of the water-soluble auxin derivative naphthaleneacetic acid (N610; Phyto-Tech Labs). The auxin-containing agar plates were prepared on the day of use from a freshly made 800mM stock solution in water and stored in the dark.

## Cisplatin sensitivity assay

Cisplatin plates were prepared as described previously [28]. L4 larvae were isolated and grown for 24h before exposure to cisplatin. After 24h cisplatin exposure, death was determined by the absence of touch-provoked movement when stimulated by harsh touch using a platinum wire.

## ASNA-1 model prediction

Three models for the *C. elegans* ASNA-1 (CELE_Zk637.5 #1–342) were obtained using the 3D modeling prediction program MODELLER [47], having as a template the structure of the Get3 protein complex of *Chaeromium thermophilum* binding AMPPNP a non-hydrolyzable ATP agonist (PDB:3IQW) [48]. Two of the models were obtained by loading the sequence with one of the following mutations: A63V and ΔH164. The third structure does not include any mutation and mimics the wild-type protein. The model was assembled as a dimer and in complex with AMPPNP, Mg and Zn mimicking the structure obtained for *Chaeromium thermophilum* (PDB:3IQW).

## Supporting information

**S1 Fig. ASNA-1 is expressed in head neurons.** Representative images of 1-day old adult worms expressing ASNA-1::mNG::AID (*syb2249*) in head neurons (white arrows). (TIFF)

**S2 Fig. bi-cistronic ASNA-1^SL2::mNeonGreen::H2B.** (A) Schematic representation of bi-cistronic ASNA-1^SL2::mNeonGreen::H2B (*syb5730)*. (B) Representative fluorescence and

differential interference contrast (DIC) images worms expressing ASNA-1^SL2::mNeon-Green::H2B (*syb5730).*
(TIFF)

**S3 Fig. Schematic representation of experimental setup for auxin-inducible ASNA-1 knockdown.** (A) 4th larval stage (L4) hermaphrodites were exposed to 1mM auxin (AUX) for 48h while they produced progeny and then removed from the plate. Their progeny remaining on the auxin-containing plates were analyzed 48 hours after removal of the mothers. Similarly handled L4 worms on non-auxin NGM plates served as a control. (B) Staged L1 hermaphrodites were exposed to 1mM auxin (AUX) for 24h, 48h, and 72h and analyzed at these time points. Similarly staged unexposed larvae on non-auxin NGM plates served as a control.
(TIFF)

**S4 Fig. Representative images of worms expressing *syb2249*; *ieSi57*.** Representative images of worms expressing *syb2249*; *ieSi57* grown on plates without (NGM) or with (NGM+AUX) 1mM auxin at specific timepoints.
(TIFF)

**S5 Fig. Representative fluorescence images of wild-type, *asna-1(ok938)*, *asna-1(ΔHis164)* and *asna-1(A63V)* adult worms.** Representative fluorescence images of wild-type, *asna-1 (ok938)*, *asna-1(ΔHis164)* and *asna-1(A63V)* adult worms expressing (A) DAF-16::GFP or (B) DAF-28::GFP. White arrows indicate coelomocyte expressing DAF-28::GFP and white triangles indicate neurons expressing DAF-28::GFP.
(TIFF)

**S6 Fig. Representative fluorescent and light microscopy pictures to visualize germline in *asna-1(+)* and *asna-1(ΔHis164)* adult animals.** (A) Representative fluorescent and light microscopy pictures of *asna-1(+)* and *asna-1(ΔHis164)* adult animals expressing *itIs37* transgene to visualize germline nuclei (red) and *itIs38* transgene to visualize germ cell membranes (green) *in vivo*. White arrows indicate vulva location and white triangles indicate spermatheca placement. (B) Representative fluorescent and light microscopy pictures of *asna-1(+)* and *asna-1(ΔHis164)* adult animals expressing *itIs37* transgene to visualize germline nuclei *in vivo*. White arrows indicate the location of the vulva for the purposes of orientation and white triangles indicate sperm in the spermatheca.
(TIFF)

**S7 Fig. Analysis of ASNA-1 million mutation point mutants.** (A) Multiple sequence alignment of ASNA-1/GET3/TRC40 comparing particular domains of the protein in 4 different species. The essential domains are highly conserved. Amino acids mentioned throughout the paper are marked in blue. (B) Confocal imaging merge of 1-day old adults co-expressing mCherry::SP12 with either ASNA-1::GFP or ASNA-1[A63V]::GFP. (C) Expression from the *hsp-4p::GFP* reporter (*zcIs4*) imaged by fluorescence microscopy in the 1-day old adult wild-type and *asna-1(A63V)* animals. *hsp-4p::GFP* expression quantification in the wild-type (n = 5) and *asna-1(A63V)* animals (n = 5). Statistical significance was determined by the independent two-sample t-test. Bars represent mean ± SD. (D) Relative mRNA analysis of ER stress reporter *hsp-4* in 1-day old adult *asna-1(A63V)* animals. Statistical significance was determined by the independent two-sample t-test. Experiments were performed in triplicate. F44B9.5 was used as a normalizing control. Bars represent mean ± SEM. Relative mRNA analysis of (E) the mitochondrial stress reporters (*hsp-6* and *hsp-60*) and (F) oxidative stress reporters (*gst-4*, *gst-30*, and *gst-38*) in 1-day old adult wild-type and *asna-1(A63V)* animals. Statistical significance was determined by the independent two-sample t-test. Experiments were performed in triplicate.

F44B9.5 was used as a normalizing control. Bars represent mean ± SEM.
(TIFF)

**S8 Fig. Full uncropped image of Western blot.** Full uncropped image of Western blot to estimate ASNA-1 levels in animals carrying a single amino acid mutation in ASNA-1. The blot was probed with an anti:ASNA-1 antibody. Tubulin was used as a loading control.
(TIFF)

**S9 Fig. Full uncropped image of Western blot.** Full uncropped blot image used in Fig 5C.
(TIFF)

**S10 Fig. Glycosylation of *asna-1(+)* and *asna-1(A63V)* animals.** Full uncropped image of Western blot following reducing SDS-PAGE to detect glycosylated (gSEC-61β) and non-glycosylated (SEC-61β) SEC-61β in strains carrying *3xFlag::SEC-61β::opsin* (*rawEx64*) transgene in *asna-1(+)* and *asna-1(A63V)* background. The dotted line represents a place of membrane cut in order to simultaneously probe with anti:Flag and anti:tubulin antibodies. Band intensity quantification of glycosylated vs non-glycosylated SEC-61β (gSEC-61β/ SEC-61β). Bars represent mean ± SD. Statistical significance was determined by the independent two-sample t-test.
(TIFF)

**S11 Fig. Full uncropped image of Western blot.** Full uncropped blot image used for quantification in Fig 6C and 6D.
(TIFF)

**S12 Fig. In silico model of *C. elegans* ASNA-1 protein.** Ribbon diagram of the ASNA-1 dimer obtained with MODELLER in its "closed conformation", bound to AMPPNP and mimicking the configuration of the Get3 protein complex of *Chaeromium thermophilum* (PDB:3IQW). The peptide chains are represented in light-orange and yellow-orange, in white the AMPPNP, in green Mg atom, in grey Zn atom, in pink the Ala63, in deep salmon the His164.
(TIFF)

**S13 Fig. Full uncropped image of Western blot.** Full uncropped blot image used for quantification in Fig 6E.
(TIFF)

**S14 Fig. Schematic model to describe function separation of the ASNA-1 protein.** Schematic model to describe function separation of the ASNA-1 protein showing the effects on redox balance (1), cisplatin resistance/sensitivity (2) and DAF-28/insulin secretion (3). A tail anchored protein (TAP) is shown in green with a cylindrical C-terminal membrane spanning domain. Normal TAP insertion is depicted with three TAP protein molecules in the ER membrane bilayer, while lower TAP insertion is shown with one such molecule. ASNA-1 protein in loss-of-function (lf) scenario is shown in red to indicate the low ASNA-1 levels that are present only because of maternal inheritance in the homozygous mutants from *asna-1/+* mothers. The altered redox balance in the point mutants (A63V and ΔHis164) resulting in lower levels of reduced ASNA-1 leads to engagement of fewer TAP protein molecules and consequently fewer TAP protein molecules inserted into the ER membrane bilayer. Created with BioRender.
(TIFF)

**S1 Table. List of million mutation project strains carrying a point mutation in ASNA-1.**
(XLSX)

**S2 Table.** *C. elegans* **strains used in this study.**
(XLSX)

**S1 Data. Data that underlies this paper.**
(DOCX)

## Acknowledgments

We thank the Caenorhabditis Genetic Center (funded by NIH Office of Research Infrastructure Programs P40 OD010440) and National Bioresource Project for the Experimental Animal "Nematode C. elegans" for providing strains.

## Author Contributions

**Conceptualization:** Dorota Raj, Gautam Kao.

**Data curation:** Dorota Raj, Gautam Kao.

**Formal analysis:** Dorota Raj, Gautam Kao.

**Funding acquisition:** Dorota Raj, Peter Naredi.

**Investigation:** Dorota Raj, Agnieszka Podraza-Farhanieh, Pablo Gallego, Gautam Kao.

**Methodology:** Dorota Raj, Gautam Kao.

**Project administration:** Dorota Raj, Gautam Kao, Peter Naredi.

**Resources:** Dorota Raj, Gautam Kao, Peter Naredi.

**Supervision:** Gautam Kao, Peter Naredi.

**Validation:** Dorota Raj, Gautam Kao.

**Visualization:** Dorota Raj, Gautam Kao.

**Writing – original draft:** Dorota Raj.

**Writing – review & editing:** Dorota Raj, Gautam Kao, Peter Naredi.

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
