## [Decision Letter · Decision Letter 0]

14 Sep 2022

Dear Gautam,

Thank you very much for submitting your Research Article entitled 'Non-overlapping requirements of ASNA-1 function for insulin secretion, cisplatin resilience, and growth revealed by genetic analysis of point mutants in C. elegans' to PLOS Genetics.

The manuscript was fully evaluated at the editorial level and by independent peer reviewers. The reviewers appreciated the attention to an important topic but identified some concerns that we ask you address in a revised manuscript. In addition to the modifications suggested by the reviewers, you may want to consider a revised title that emphasizes the distinction between this work and your previously published work, such as "Determinants of non-overlapping functions...".

We therefore ask you to modify the manuscript according to the review recommendations. Your revisions should address the specific points made by each reviewer.

[LINK]

Yours sincerely,

Cathy Savage-Dunn

Guest Editor

PLOS Genetics

Gregory P. Copenhaver

Editor-in-Chief

PLOS Genetics

Reviewer's Responses to Questions

**Comments to the Authors:**

Reviewer #1: In the manuscript by Raj et al, the authors use C. elegans to characterize phenotypic effects of different mutations in the essential cytosolic ATPase, ASNA-1 gene. They also examine phenotypes resulting from ASNA-1 being expressed in different tissues. Interestingly, they find that tissue specific expression of ASNA-1 results in different phenotypes – and the same is the case for different ASNA-1 mutations affecting evolutionary conserved amino acids.

This is interesting because it suggests that ASNA-1 can be targeted pharmacologically with a view of specifically influencing certain phenotypes but not others. ASNA-1 is a biomarker for many different diseases. Since ASNA-1 plays an important role in cisplatin resistance and insulin secretion, such targeted intervention has a great potential for example for improved chemotherapy with less side effects.

The paper is well written and easy to read. The data is clearly presented and the rationale underlying the experimental designs are easy to follow. Appropriate controls are included in all experiments.

Minor issues:

In the abstract and in the summary paragraph before results section the performed experiment are mentioned – but what they actually show is not revealed – why not describe the results of the experiments at this point?

It would be nice to have a better introduction/description of the Switch I and II domains in the beginning of the paper.

Auxin induced degron (AID) – write full first time it is introduced

I find it a bit odd that the results section starts with three supplemental figures – it kind of suggests to me that perhaps the order of the first results could be changed a bit?

The statement “significant delay in development” – lacks proper statistical testing. I am sure the phenotype is fairly clear but without quantification it is an unfortunate choice of word. Growth rate, size, time to first egg lay could easily be measured to allow quantification and statistical testing

Structure and function of human proteins – should be plural?

…we used the transgenes… - clarify which transgenes?

…close contact – rather than closed contact?

Figure 1 – is the panel with DIC NGM+AUX omitted on purpose?

It is not clear if the data in Figure 3A and B is just one experiment or a representative example. The results are very clear – but some quantification and statistics would perhaps be appropriate. Same for Figure 4G and H.

…the mutant had no effect on… should be the mutation had no effect on?

Defect in subcellular – or defective subcellular ?

I am a little confused about the use of present and past tense in the results section – I think sticking to one makes the paper easier to read. Common practice for the journal should be followed.

I appreciate that the authors very nicely show that mutations can target only one part (and hence a specific phenotype) of ASNA-1 leaving the other part unaffected. Indeed, as the authors suggest, a similar effect can perhaps be seen with a drug. Perhaps the authors could elaborate on what kind of drug the envision could be used to target fx the Switch I and II domains – and what are their thoughts on doing that without influencing the entire folding of the ASNA-1 protein?

Figure S9 Species rather than spices

Reviewer #2: The review has been uploaded as an MS Word document

Reviewer #3: Manuscript Number: PGENETICS-D-22-00914

Title:Non-overlapping requirements of ASNA-1 function for insulin secretion, cisplatin resilience, and growth revealed by genetic analysis of point mutants in C. elegans

Authors: Dorota Raj, Agnieszka Podraza-Farhanieh, Pablo Gallego, Gautam Kao and Peter

Naredi

This study is focused on the functional effect of two mutations on the metalloregulated ATPase ASNA-1. ASNA-1 function has already been described and has been related with insulin secretion and cisplatin sensibility in mammalian cells and in C. elegans. The authors analyzed the consequences of specific two-point mutants that separate ASNA-1 cisplatin response function from its role on insulin secretion. They used the C. elegans model as part of their experimental strategy and showed the tissue requirements of ASNA-1 for C. elegans growth and development. The authors also showed that targeting single residues in ASNA-1 affecting the Switch I/II domain function, in comparison to the complete knockdown, decreases cisplatin resistance without compromising other important biological functions.

This study is in the logical next step from the authors’ previous findings, showing that at least two residues can modify ASNA-1 biological function with respect to cisplatin sensitivity.

Nevertheless, in my opinion there are a number of issues that need to be addressed before this paper can be accepted for publication in PLOS Genetics.

1) There is too much relevant evidence in the supplementary figures that makes the paper difficult to follow. In fact, the Results start with supplementary instead with Figure 1, Perhaps the section “ASNA-1 is broadly expressed in C. elegans”, could be reorganized as figure 1.

2) Figure 1. “Soma-specific ASNA-1 depletion leads to L1 larval arrest” must include images (transfer from supplementary figure) and quantification of L1 vs control.

3) Figure 2. Intestine-specific ASNA-1 depletion leads to developmental delay. The figure must include the quantitative analysis of the individuals with developmental delay vs control.

4) Figure 3. The figure must include representative images for each graph “cytoplasmic localized DAF-16::GFP (n ≥ 15) and adults with secreted DAF-28::GFP in coelomocytes (n ≥ 20). They must include the number of independent assays and statistical analyses must be done.

5) Figure 4. The figure must include representative images for graphs G and H. They must include the number of independent assays and statistical analyses must be done

6) The discussion needs to emphasize the relevance of the findings in the context of cisplatin resistance in cancer therapy.

**Have all data underlying the figures and results presented in the manuscript been provided?**

Reviewer #1: **No: **It seems that all data has not been included.

Reviewer #2: Yes

Reviewer #3: Yes

PLOS authors have the option to publish the peer review history of their article (what does this mean?). If published, this will include your full peer review and any attached files.

Reviewer #1: No

Reviewer #2: **Yes: **Keith Nehrke

Reviewer #3: No

---

## [Decision Letter · Decision Letter 1]

21 Nov 2022

Dear Gautam,

We are pleased to inform you that your manuscript entitled "Identification of C. elegans ASNA-1 domains and tissue requirements that differentially influence platinum sensitivity and growth control" has been editorially accepted for publication in PLOS Genetics. Congratulations!

Yours sincerely,

Cathy

Cathy Savage-Dunn

Guest Editor

PLOS Genetics

Gregory P. Copenhaver

Editor-in-Chief

PLOS Genetics

Comments from the reviewers (if applicable):

Reviewer's Responses to Questions

**Comments to the Authors:**

Reviewer #1: I find that the revised manuscript is much improved and ready for publication.

Reviewer #2: The authors were responsive to the critiques and have addressed most of my comments.

Reviewer #3: The authors have addressed the majority of my concerns and their paper is much improved. I recommend publication

**Have all data underlying the figures and results presented in the manuscript been provided?**

Reviewer #1: Yes

Reviewer #2: Yes

Reviewer #3: Yes

PLOS authors have the option to publish the peer review history of their article (what does this mean?). If published, this will include your full peer review and any attached files.

Reviewer #1: No

Reviewer #2: **Yes: **Keith Nehrke

Reviewer #3: No

**Data Deposition**

http://datadryad.org/submit?journalID=pgenetics&manu=PGENETICS-D-22-00914R1

**Press Queries**

---

## [Editor Report · Acceptance letter]

5 Dec 2022

PGENETICS-D-22-00914R1 

Identification of *C. elegans* ASNA-1 domains and tissue requirements that differentially influence platinum sensitivity and growth control 

Dear Dr Kao, 

We are pleased to inform you that your manuscript entitled "Identification of *C. elegans* ASNA-1 domains and tissue requirements that differentially influence platinum sensitivity and growth control" has been formally accepted for publication in PLOS Genetics! Your manuscript is now with our production department and you will be notified of the publication date in due course.

With kind regards,

Anita Estes

PLOS Genetics

On behalf of:
